# Two-Step Floating Catchment Area Model-Based Evaluation of Community Care Facilities’ Spatial Accessibility in Xi’an, China

**DOI:** 10.3390/ijerph17145086

**Published:** 2020-07-14

**Authors:** Sunwei Liu, Yupeng Wang, Dian Zhou, Yitong Kang

**Affiliations:** School of Human Settlement and Civil Engineering, Xi’an Jiaotong University, Xi’an 710049, China; wang-yupeng@mail.xjtu.edu.cn (Y.W.); kyt2013@stu.xjtu.edu.cn (Y.K.)

**Keywords:** GIS, 2SFCA model, potential model, community care facilities, spatial accessibility

## Abstract

Due to the rapid increase in the number of elderly people in Chinese cities, the development and planning of aged care facilities, and particularly community care facilities, which will gradually become the mainstream choice for the elderly in China, is becoming an important topic for urban sustainability. Previous studies have shown that the number and scale of aged care facilities in many cities are far from meeting the needs of the elderly and the overall occupation rate is low. Some of these cities are still expanding and some are undergoing urban renovation. In this process, the scientific planning of community care facilities to promote efficient use of facility resources has become an urgent problem that needs to be solved. In this study, the two-step floating catchment area (2SFCA) method and a potential model based on the Geographic Information System (GIS) were used to carry out a scientific evaluation of the spatial accessibility of community care facilities in the Beilin district of Xi’an. The aims were to explore the best quantitative research methods for assessing the distribution of Xi’an community care facilities’ spatial accessibility, provide ideas for similar studies in the future, and further the understanding of spatial allocation of urban community care facilities resources.

## 1. Introduction

Progressive rises in urban population accompanied by the aging of societies are two trends in cities globally, and particularly in developing countries [1,2]. To a certain extent, population aging is a sign of the progress and development of social civilization, but it also brings great pressure to social economic security and medical care [3,4]. A scientifically reasonable configuration of community care facilities in a city can alleviate problems associated with their spatial distribution and improve their efficiency. For current urban planning and architecture, there is an urgent need to explore this new topic. Thus, application of scientific methods to evaluate the spatial accessibility of Xi’an care facilities can lead to reasonable space configurations and has important practical significance. Two models were used for this evaluation and compared in this study.

### 1.1. Aging Cities

Aging is essentially a demographic process grounded in demographic transition theory [5,6]. Population ageing is taking place across all countries of the world [7]. According to the UN World Population Prospects 2017 [8], there were 962 million people aged 60 and over in 2017, which is more than double the number in 1980 (382 million). The proportion of the global population aged 65 and over rose from 5% in 1960 to 9% in 2018 and it is expected to rise to 16% by 2050, when nearly 80% of the older population will be living in developing countries [9]. The rapid growth of the elderly population has become a prominent and global phenomenon, including in developed countries such as Canada [10] and Japan [11] and developing countries such as Thailand [2] and Romania [12]. The rate of population aging is a concern of most nations, manifesting as critical issues in public health, preventive medicine, and socio-economic sectors. To engage and assist cities to become more “age-friendly,” the World Health Organization (WHO) prepared the Global Age-Friendly Cities Guide and a companion “Checklist of Essential Features of Age-Friendly Cities” [13], which represents a significant effort to adjust the focus of urban planning to this aging situation.

By the end of 2018, there were 249 million people aged 60 or above in China. By 2030, this number is expected to grow to 358 million, accounting for 25.1% of China’s total population. By 2050, there will be 487 million people, or 35.1% of the total population, aged 60 or above. In 2050, it is expected that there will be 2.02 billion people over the age of 60 globally, with China accounting for 24.1% of this global total [14].

### 1.2. Community Care Facilities

In recent years, policy recommendations to “rebalance” health systems have intensified [15,16], seeking to respond to the growing, and more complex, health and social care needs of aging populations [17], thus revealing a shift from hospital to community and primary care [18]. Aged care facilities are usually divided into home and community care (HACC) and residential aged care (RAC) in international research [19,20]. In Sweden, an increasing number and proportion of older people with extensive care needs are living in the community [20]. In the Netherlands, older persons have begun using community-based care more often [21]. The new policy of Taiwan has focused on providing home- and community-based services [22]. It is reported that social participation is beneficial for the postponement of frailty of the elderly [23,24]. The research of Kim J. (2020) and Joan K. (2016) also emphasized the importance of the community for active aging among older adults [25,26]. Improving accessibility to community care facilities is also beneficial for promoting regular screening of some diseases, such as diabetic retinopathy [27]. It can be seen that home and community aged care is a trend worldwide and the desire to “rebalance” puts pressure on community care facilities in various countries.

Aged care facilities in China mainly consider two types of institutional care facilities and community care facilities [28,29]. Institutional care facilities for the elderly include nursing homes and elderly apartments, which are comprehensive facilities for providing the elderly with daily care, rehabilitation nursing, medical care, and other services. Community care facilities mainly refer to community healthcare, elderly activity centers, etc., where government agencies or social organizations provide home-based elderly people with life care, rehabilitation training, cultural activities, and other services based on the communities. The elderly usually only use these facilities during the day when they are needed and there are corresponding staff in the facilities to provide the services. The number of beds is an important indicator of the ability of elderly care facilities to provide physiological or medical care. In this paper, the information on the number of beds and coordinates of community care facilities comes from the Xi’an Planning Bureau. As China has become an aging society sooner than expected, the number of beds in aged care facilities is insufficient [30,31]. Due to cultural factors, the elderly are also more inclined to choose home- and community-based care, which means greater pressure on community care facilities. Scientific planning of community care facilities to maximize their utilization efficiency in cities is a significant step toward coping with this stress and reducing social costs.

### 1.3. Theoretical Model on Research of Urban Aged Care Facility Planning

Research on urban aged care facilities in recent years has mainly focused on the discussion of old-age care models and policies [12,32,33,34,35,36,37,38] and investigation of the living needs of the elderly and the environment [39,40,41,42,43,44]. Most of these studies use observation, questionnaires, and interview methods, amongst other approaches, to obtain data. Some pay more attention to the influencing factors and correlation mechanisms of facility planning and some explore quantitative methods and strategies. For example, Tsukahara, K. et al., (2019) developed a method to evaluate the location of aged care facilities, from the viewpoint of whether they are equitably located for users, using the improved Median Share Ratio (MSR) [45].

In recent years, optimization methods have been applied to the research of urban facility planning [45,46,47] and the Geographic Information System (GIS) approach has been widely used [11,48,49]. Nishino, T. (2017) explored the quantitative properties of the macro supply and demand structure for facilities for the elderly who require support or long-term care throughout Japan and they presented these properties as an index value [47]. Higgs, G. et al. (2019) evaluated the potential access to primary healthcare services in Wales using GIS-based tools to examine variations in geographical accessibility to general practitioner (GP) surgeries offering appointment times outside of “core” operating hours [49]. The two-step floating catchment area (2SFCA) model and the potential model are also widely used in spatial accessibility evaluations, especially of medical resources [50,51,52,53,54,55,56]. Tao Z.L. used a potential model to evaluate the spatial accessibility of healthcare facilities in Beijing [56]. Use of the 2SFCA model has also gradually expanded in other fields, such as food supply [57], urban fire service [58], public daycare and kindergartens [59], and aged care facilities. Tseng et al. (2018) applied spatial analysis techniques, such as hot-spot analysis and the enhanced two-step floating catchment area (E2SFCA) method, to examine Taiwan’s distribution of the aging population of each village [60]. The 2SFCA model is also constantly being studied [61,62,63]. The research of Chen X et al. (2019) shows that on a small scale analysis (e.g., the community level), the catchment size is the most critical model component; on a large scale analysis (e.g., statewide), the distance decay function is of elevated importance [61]. 

Large areas, usually on the city or urban area scale, have been selected in most studies using the 2SFCA method. Therefore, the problems and solutions obtained through analysis are not detailed and targeted. The area of the present study comprises four sub-districts, allowing a focus on the problems related to community elderly care facilities in the study area and the proposal of more targeted improvement strategies. Furthermore, previous research has generally used a single model, which could be defective and not accurate, or has highlighted the comparative characteristics of the model. In this study, the 2SFCA model was combined with a potential analysis model to obtain more accurate and scientific results.

### 1.4. Acceptable Travel Distance for the Elderly

In Western countries [64,65], the elderly are more dependent on cars, but 80% of trips less than 0.5 km involve walking [66]. This is a different situation from developing countries, in which the elderly mainly choose public transport and walking [67,68,69]. Some residents in the center of Oslo describe walking as an ideal way to travel to destinations that are not too far away (such as gyms and healthcare facilities) [70]. In Greater Rotterdam, the Netherlands, walking seems more important for the elderly than non-elderly people; for both the elderly and non-elderly, walking is more likely preferred for short distances [71]. Therefore, it is considered that elderly people will arrive on foot when using community care facilities that are near their home.

Some studies have shown that the generally acceptable walking distance for the elderly is between 500 and 700 m [72,73]. A study of the elderly in the Netherlands showed that most trips to grocery stores exceeded 700 m on foot and even the majority of elderly people with functional limitations traveled more than 500 m [72]. In a study in Lisbon, 10- and 15-min walk thresholds were chosen by asking 30 elderly people about what an acceptable maximum duration would be for them to walk to the nearest pharmacy. At a more inclusive speed of 0.8 m/s, an older individual would walk 480 m in 10 min and 720 m in 15 min [73]. According to GPS watches of a sample of 30 elderly people in Xi’an city, behavioral trajectory analysis showed that when the destination is restricted by the location of facilities and residences, the travel distance of the elderly does not exceed 800 m; when there is an opportunity to choose an activity space, the travel distance will be shortened and mostly not exceed 600 m [74].

Above all, due to the global trend of an aging population and the concept of home- and community-based care services, it is necessary to evaluate the accessibility of community care facilities. As in many countries, especially developing countries, the urban structure is constantly changing and the processes of data collection, accessibility evaluation, and scientific planning of community care facilities will continue to be repeated for a long time in the future. Related research about the planning and evaluation methods will provide an important reference for policymakers and planners. 

Planning methods of institutional and medical facilities are well studied in previous research [43,45,46,47,48,49,50,51,52,53,54,55,56]. This study focuses on community care facilities, which have different service radii and characteristics compared to the other care facilities. Meanwhile, this research includes comprehensive results of different quantitative methods and ranks the accessibility of the facilities for selected communities, which provides a more intuitive reference for planners.

## 2. Methodology

### 2.1. Selected Area

By 2010, the number of people aged 65 and older in Xi’an was more than 0.71 million, reaching 8.5% of the total population, growing from 1.99% in 2000. By 2015, it increased by a further 2.04% compared with 2010 [14]. The population density distribution of the elderly also indicates that Xi’an is aging at an accelerating rate [75]. A total of 49 communities in the four sub-districts of Baishulin, Dongguannanjie, Wenyilu, and Taiyilu of the Beilin district, Xi’an were selected in this study to evaluate the spatial accessibility of their community care facilities. The boundaries and names of the communities are shown in Figure 1. There was a total of 32 community care facilities. 

### 2.2. 2SFCA Model

The 2SFCA model, based on an improvement of the early floating catchment area model, is an important method in the study of public facilities’ spatial accessibility. It has been widely applied and used to develop many improved models in domestic and foreign research. In this method, a threshold was set (an acceptable maximum duration for travel time or distance) and supply or demand points (residence point) were only calculated within that threshold, regardless of the characteristics of the city outside the threshold. Thus, the model’s focus is on the principle of proximity when there is little difference in the conditions of each facility in the city (the principle is shown in Figure 2).

The first step is to search for the elderly population locations (*k*) within the distance of *d*_0_ (threshold), centering on any activity facility location *j*, and calculate the supply–demand ratio *R_j_* of each facility:(1)$$R_{j}=S_{j}/\sum_{k\in \{d_{ij}\leq d0\}}D_{k},$$
where

*R_j_*—ratio of beds of facility location *j* and elderly population in search area *i* (*d_ij_* ≤ *d*_0_);*S_j_*—beds of facility location *j* (the total supply at point *j*);*D_k_*—demand for beds for the aged within the threshold (*D_kj_* ≤ *d*_0_);*d_kj_*—the extreme travel distance of the elderly between positions *k* and *j*; and*d*_0_—threshold.

In the second step, for each settlement *i* of the elderly, the number of facilities within the threshold (*d*_0_) is determined and the ratios of supply and demand (*R_j_*) of all facilities are summed together to obtain the spatial accessibility of settlement i:(2)$$A_{i}=\sum_{j\in \{d_{ij}\leq d0\}}R_{j}=\sum_{j\in \{d_{ij}\leq d0\}}\left[S_{j}/\sum_{k\in \{d_{ij}\leq d0\}}D_{k}\right].$$

Since the search process only focuses on the relative position between population and facilities, an important advantage of the 2SFCA model is that it considers the usage of facilities across administrative boundaries. That is, the internal boundaries of the community have no influence on the process and results of the 2SFCA calculation. The outer boundary of the entire study area will have a very limited impact on the results of the entire area because relatively few results are related to the outer boundary.

As a tool for geographic data collection, storage, and analysis, GIS shows geography information as well as information in other disciplines, such as urban population, social resources, and economics. Because of its analysis capabilities, GIS has unique advantages for spatial accessibility evaluations based on the 2SFCA model:“Feature to Point”—This function calculates the community geometric center instead of the population distribution center;“Proximity”—This function is used to search for elements based on the community’s population center and facilities; and“Join” and “Calculation”—These functions are used for correlation calculations between different data.

As shown in Figure 3, the 2SFCA model can be easily implemented in GIS.

### 2.3. Potential Model

The potential model is one of the classical models of spatial interaction. The potential represents the energy generated by an object with respect to another object. For example, the energy *A_i_*_j_ generated by *j* to *i* is *M*_j_/*D_i_*_j_, where *M_j_* represents the scale of activity of point *j* and *D_ij_* represents the travel impedance factor (on distance or time) between point *i* and point *j*. The potential of all objects in a system to a point is equal to the sum of the potential generated by each object at that point. For instance, if there are n discretely distributed objects in space and the travel friction coefficient is β, then the total potential Ai for point *i* is
(3)$$A_{i}=\sum_{j=1}^{n}A_{ij}=\sum_{j=1}^{n}\frac{M_{j}}{D_{ij}^{\beta }}.$$

The potential model is similar to the 2SFCA model in principle. Both of them require the GIS-based “Proximity” function to perform two searches, centered on facilities and the elderly population. However, there is no determined search radius in the potential model, but it attenuates the calculated supply–demand ratio based on the distance of the facility to the elderly. Its implementation in GIS is shown in Figure 4.

### 2.4. Comparison of the Two Models

The above two research methods are common methods for evaluating spatial accessibility. The main difference is that the potential model pays more attention to the distance attenuation effect of spatial distance in accessibility. In the 2SFCA model, the actual distance between the center of gravity of the elderly population distribution and the facilities within the search radius does not affect the spatial accessibility results, that is, facilities in the distance and facilities in the vicinity have the same opportunity for the elderly to be selected. In the potential model, under the same conditions, the spatial accessibility of facilities with longer distances is lower than that with shorter distances and the greater the distance, the more the accessibility of the facility decreases. These two methods are used to evaluate the spatial accessibility of community care facilities in Xi’an. The results are compared to find more suitable and scientific methods and conclusions.

### 2.5. Data Preparation

In this study, the number of beds in the community care facilities was used as the service supply for the elderly in the selected area. Because of the small scale of the communities, in order to simplify the model, the geometric center of each community replaced the center of the elderly population in the center of the second search domain in the 2SFCA method [76]. The data of the elderly population were collected by the government of Beilin District in Xi’an in 2015 and the population of people over 70 years old was used in this research.

This study carried out two calculations with search radii of 500 and 600 m using the 2SFCA method. The basis was as follows: (1) As introduced in the introduction section, according to the walking distance of the elderly in many countries, as well as the tortuousness and diversity of their actual walking behavior, and allowing for their walking speed to be related to distance attenuation (the search radius as a straight line distance must be less than the actual walking distance), the walking range of elderly pedestrians was taken to be 500 m. (2) According to samples of the elderly in Xi’an, most of their single travel distances were between 600 and 800 m. For the same reason, 600 m was used as another search radius.

In the potential model, the linear distance between two points instead of the actual road network distance was used as the D value. In actual operation, the travel friction coefficient β is mostly concentrated between 1 and 2 and is most commonly 1 or 2 [77]. In this study, the calculation results of the travel friction coefficients of 1 and 2 were compared. It was found that the results were too discrete to observe when the value was 2. Therefore, 1 was finally selected as the value of the friction coefficient in this study.

## 3. Results

### 3.1. 2SFCA Model

The 2SFCA method demonstrates the differences in the spatial configuration of the community care facilities within the scope of this study. On one hand, the smaller the size of the census unit, the more detailed the population data used. On the other hand, when the search domain moves from one census unit to another, it can cross the boundaries of the administrative area and the potential interaction between the elderly and the retirement facilities is better considered. The spatial accessibility calculations were performed on search radii of 500 m and 600 m. The results are shown in Figure 5 (the results are averaged and divided into six levels, from the minimum to the maximum value).

First, the spatial accessibility of community care facilities in the study area was shown to be considerably insufficient. According to the “Thirteenth Five-Year Plan”, every 1000 elderly people in China should have 35 to 40 beds, that is, the ratio of supply and demand should not be less than 0.035. However, the average value was 0.0031 when the search radius was 500 m and 0.0109 when it was 600 m, due to an increase in the number of searchable facilities. When the search radius was 600 m, only six communities (12%) had facilities that may meet the demand.

Second, the spatial accessibility of community care facilities in the study area was unevenly distributed. When the search radius was 500 m, the best accessibility value was 0.01357, which was in the Huzhu community. The worst community with a non-zero value of 0.00085 was Wenbei1 community; its value was just 6% of the best community. There were 11 communities (22%) with 0 accessibility. When the search radius was 600 m, the community with the best accessibility (0.04418) was also Huzhu. The worst community with a non-zero value, of 0.00108, was Le1; its value was only 2% of the best community. There were eight communities (16%) with 0 accessibility.

It can be seen from Figure 5 that when the search radius was expanded from 500 m to 600 m, the spatial accessibility of community care facilities in the study area generally increased due to the increased number of facilities that could be found in each community. In some communities where the accessibility was 0, such as Dongdajie, Xingqinggong, Tieluju, etc., when the search radius increased, the accessibility value increased. Therefore, a radius of 600 m can better show the differences between communities and is more conducive to the proposal of precise strategies. 

### 3.2. Potential Model

The calculation results of the potential model were similar to those of the 2SFCA model, as shown in Figure 6 (the results are averaged and divided into five levels, from the minimum to the maximum value).

There was no community with an accessibility value more than 0.035 (i.e., meeting the demand). The average accessibility value was 0.0022 and the number of communities with zero accessibility to the facilities increased to 26 (53%), which was significantly greater than that calculated with the 2SFCA model. The spatial accessibility of Huzhu community in Dongguannanjie was the best, with a value of 0.01609. The worst accessibility in a community with a non-zero value was that of Jingjiulu, of which its value was very close to zero.

### 3.3. Strategies

According to the proportion of the number of communities with different spatial accessibility, 49 communities were divided into four accessibility levels: accessibility accounting for the top 10% of the three simulation results (there were slight differences between the three models); 10–27% communities; 27–45% communities; and the other 55% of communities. Based on these four levels, the results of the three simulations were considered comprehensively to obtain a map of areas in the community that lack facilities, that is, a priority map for building new facilities (Figure 7). For instance, the spatial accessibility of 10 communities, such as Sanxuejie and Huzhu, in one or more model results entered the top 10%, so when adding facilities, these communities do not need to be considered. The spatial accessibility values of Hebei, Wenbei2, Yanbei, Andongjie, Huojulu, and Wenyinanlu communities were all 0 in all models. Therefore, they are the top priority when adding new community care facilities. Jiaoda community also had zero accessibility in all three results; however, no additional facilities are needed in Jiaoda and Xingqinggong communities because few elderly people live here due to the presence of the college and park. Furthermore, because of the small scale of communities, it is possible for the elderly to use facilities across the community. In areas composed of communities with low accessibility, priority should be given to new facilities in communities located in the geographic center of this area, such as Yanbei community.

### 3.4. Comparison of Models

In assessing the spatial accessibility of the community care facilities, the results of the 2SFCA method with a radius of 600 m were the highest and those of the 2SFCA method with a radius of 500 m were in the middle. The potential model had the lowest, and significantly different, calculation results. The possible reasons for this are as follows: first, the potential model had no radius limitation, so it was not accurate enough to capture the activities of the elderly who have a small range of activities. On the contrary, the 2SFCA model was more suitable for the study of elderly people engaged in nearby and similar activities. Second, due to the existence of the distance attenuation coefficient in the potential model, the accessibility calculation results quickly decayed with the increase in distance from the center of the community to the facilities. It was found that the number of communities with zero accessibility accounted for more than half of the total research area, which was not consistent with the actual situation. This was because the road network structure at the sub-district scale was relatively simple. In the case of short-distance walking, accessibility was less affected by the spatial distance and the attenuation effect with the increase of the spatial distance was not as obvious as assumed in the model. Therefore, the 2SFCA model was a better method to study the spatial accessibility of community care facilities at the sub-district and community scales compared to the potential model, both in principle and actual performance.

From the perspective of expression, the 2SFCA model with a 600 m search radius could better capture the differences between the data and thus, show the differences of each community and provide more precise evidence for analysis. The differences between the data obtained by the potential model were not appreciably observable. The results of the 2SFCA model with a 500 m search radius were better than those of the potential model and worse than those of the 2SFCA model with a radius of 600 m in terms of expression. Furthermore, 600 m was found in the actual sample survey, which is closer to the real travel distance of the elderly in Xi’an.

## 4. Discussion

From this study, the calculation results of different models all showed that the accessibility of community care facilities in 49 communities in the four sub-districts of Baishulin, Dongguannanjie, Wenyilu, and Taiyilu in Beilin District were generally insufficient. The accessibility value of most communities was less than 0.035 (i.e., the number of beds per thousand elderly people was less than 35), of which the average value calculated by the potential model was 0.0022, the average value calculated by the 2SFCA model with a radius of 500 m was 0.0031, and the value was 0.0109 when the radius was 600 m. On the other hand, the accessibility of community care facilities was not uniform within the study area. For example, in the 2SFCA model with a radius of 600 m, the community with the best accessibility had a value of 0.0442, which exceeded the standard of 35–40 beds per thousand elderly people in the 13th Five-Year Plan; however, the accessibility of facilities in eight communities was zero and these were scattered across the study area. The reasons may be as follows. First, the aging population in Xi’an has grown quickly and there are generally insufficient aged care facilities in the entire urban area. Second, there are many residential communities, especially old ones, so there is a large number of elderly people. In addition, the research area was developed early and public facility systems such as aged care facilities were relatively stable before the issue of aging was valued, making it difficult to renovate.

This study integrates results of the three models and divides the accessibility of community care facilities into four levels. According to the accessibility level of each community, the communities are recommended to be divided into different priority levels when adding new care facilities or beds. Among these, 7 communities such as Hebei had the highest priority, 10 communities such as Duanlvmen had high priority, 9 communities such as Machangzi had low priority, 10 communities such as Juhuayuan had very low priority, and 11 communities such as Sanxuejie were not considered to require new facilities or beds at this time.

In general, research using the 2SFCA model based on GIS has positive theoretical and practical significance for evaluating spatial accessibility and planning strategies of community care facilities. On one hand, there are many studies on China’s aged care services that focus on the choice of care model, the construction of the care service system, and residents’ choice of facilities. They have often focused on the influence of traditional Chinese concepts on the choice of care models, especially residential care facilities. There are few studies on the spatial accessibility of community care facilities, but this area is expected to expand in future development worldwide. On the other hand, the methods used in previous related research are mainly descriptive analyses and interview surveys. The present study used the GIS-based 2SFCA method combined with the potential model and calculated the results under different search radii, considering the use of facilities across communities and sub-districts. The combined results of the three calculations were used to make recommendations for the planning of community care facilities. In the future, the project can be improved by further refining the location center of the elderly population, which was replaced by the community center in this research.

Previous studies have examined the accuracy and sensitivity of the 2SFCA model [78,79,80,81]. The results of the E2SFCA method proposed by Kanuganti et al. were compared statistically with the observed accessibility values collected via a questionnaire survey and an acceptable absolute percentage error (<10%) was found in all the districts [78]. The results of the 2SFCA model and its different improved models can be different, but they all exhibited a strong mutual correlation [79,80]. The research by Luo et al. in 2012 involved both rural and urban areas, which performed oppositely in the sensitivity test on population threshold, but the accessibility changes of different thresholds were consistent [81].

### 4.1. Model Shortcomings and Improvements

The 2SFCA method is demonstrated to have two fundamental shortcomings as proved by McGrail and Humphreys. First, it uses only one catchment size for all populations. Second, it assumes undifferentiated proximity within a catchment (which is especially problematic when the catchment is large) [82]. In response to these defects, researchers have proposed various improvements to overcome the distance decay problems, extend the search radius, quantify supplies or demands, and consider multiple travel patterns [79,83,84]. McGrail and Humphreys proposed a Dynamic 2SFCA method (D2SFCA) in 2014, which set different search radii according to the population density of regions [81]. Kanuganti et al. proposed an enhanced 2SFCA (E2SFCA) method for finding accessibility to healthcare in rural areas by calibrating a distance decay function [78]. Fransen et al. put forward a commuter-based version of the 2SFCA method to account for daily mobility [80]. Unlike the 2SFCA method, the potential model mainly improved via the introduction of Vj (the impact of population size) decades ago [85,86] and later research has continued to use this improved method [87]. 

The target populations in most research are complex and a variety of travel modes can be selected, so an obvious deviation can be caused by this first shortcoming. In this study, the 2SFCA method with radii of 500 m and 600 m was used, based on the minimum acceptable walking distance for the elderly. Most elderly people walk on foot when using community care facilities and the acceptable walking distance for them is not much different, so there is a relatively small deviation. Moreover, the results of the potential model show that the accessibility for more than half of the communities in the selected area was 0, where the distance attenuation effect was overestimated. This may be because within an acceptable walking distance, the elderly tend to go to “available” facilities rather than “closer” facilities. That is to say that distance is not a determinant factor affecting the elderly to choose community care facilities. This is very different from traveling farther away.

### 4.2. Policy Implications

According to statistics, China’s urbanization rate exceeded 60% in 2019 [88] and is expected to reach 70% by 2030 [89]. With the rapid increase in the size of the urban population, the construction of new districts based on urban expansion and the renovation of old urban areas have become important means of urbanization. The spatial structure characteristics of the interior and exterior of the city are undergoing rapid and tremendous changes. This not only provides space to re-plan elderly facilities, but the high speed of urban development and the lag of resources also creates challenges in the reasonable layout of facilities. The GIS-based 2SFCA method and potential model are suitable approaches to evaluate the accessibility of care facilities in smaller areas (such as communities and sub-districts) and in larger areas (such as districts and cities) due to the advantage of crossing administrative boundaries. These can play a guiding role in the arrangement of aged care resources in the expansion and renovation processes of various cities. 

In 2015, the results of a population survey conducted by the Xi’an Statistics Bureau showed that the city’s elderly population over 60 years old reached 1.352 million, accounting for 15.53% of the total population. It is expected to reach its peak in 2040, when the proportion will reach 30% [14]. Many cities around the world are following a similar trend as Xi’an. More than one model was run in this study to ascertain a more complete image of the accessibility landscape of the study area. Using this method, the spatial accessibility of community care facilities in various countries or regions can be predicted in the future according to demographic projections of the aging process. This will help planners to plan ahead for care facility resources, which has important policy implications.

## 5. Conclusions

The high rate of growth of the aging population leads to a high demand for optimized community facility planning for elderly people. The 2SCFA and potential models to evaluate community facility accessibility were applied and compared in Beilin District, Xi’an, China. The results demonstrated that the potential model was more suitable to study accessibility over a large range with a longer travel distance because it did not have the limitations of a search radius or require the introduction of a travel friction coefficient. In contrast, the 2SCFA model was more suitable for short-distance walking scenarios because it represented the actual usage of community care facilities. Furthermore, the 2SCFA model with a search radius of 600 m can better show the differences of various communities, which is conducive to priority planning of new facilities. This evaluation approach using the 2SCFA model for community facilities could be adopted in the processes of urban development and redevelopment and it also provides a reference for similar problems in other countries and cities.

Limited by access to demographic data, replacing the population center with the community center may have led to inaccurate results in this study. The location of the elderly population can be further refined by detailed investigations in future research and similar community-scale studies. This method is reproducible and due to the intensification of global aging and the continuous changes to urban structure, this evaluation method has important policy guidance significance.

## Figures and Tables

**Figure 1 ijerph-17-05086-f001:**
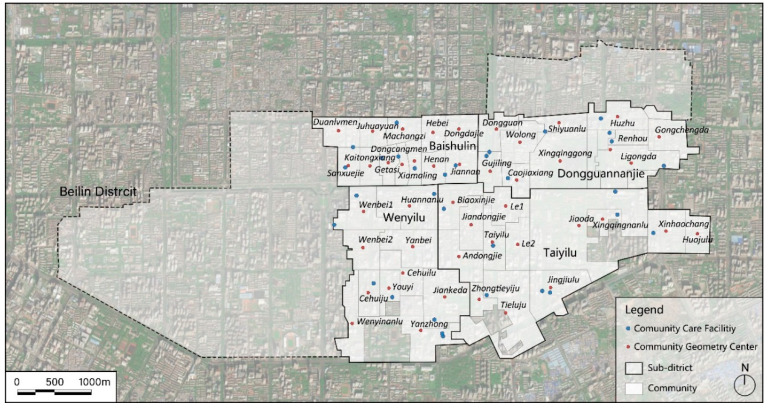
The distribution of community care facilities in the study area. Also shown are sub-district boundaries. Italicized font labels are community names and regular font labels are sub-districts.

**Figure 2 ijerph-17-05086-f002:**
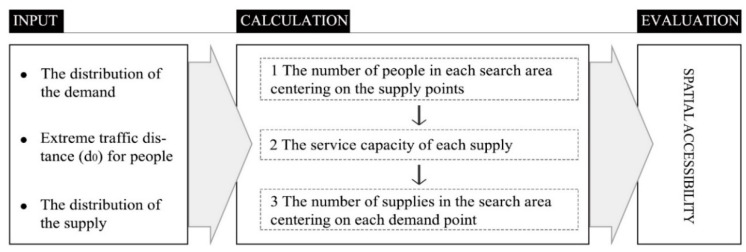
Principle of the two-step floating catchment area (2SFCA) model.

**Figure 3 ijerph-17-05086-f003:**
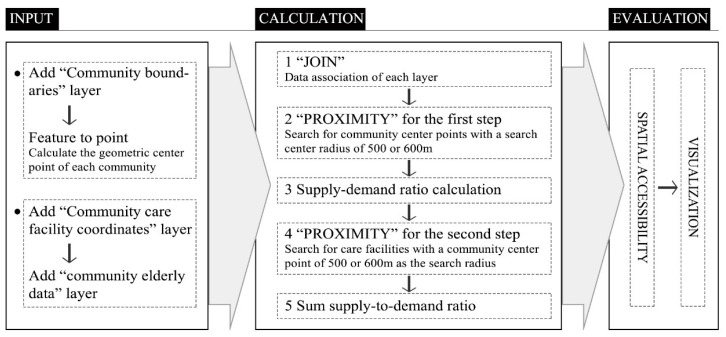
Implementation of the 2SFCA model in the Geographic Information System (GIS).

**Figure 4 ijerph-17-05086-f004:**
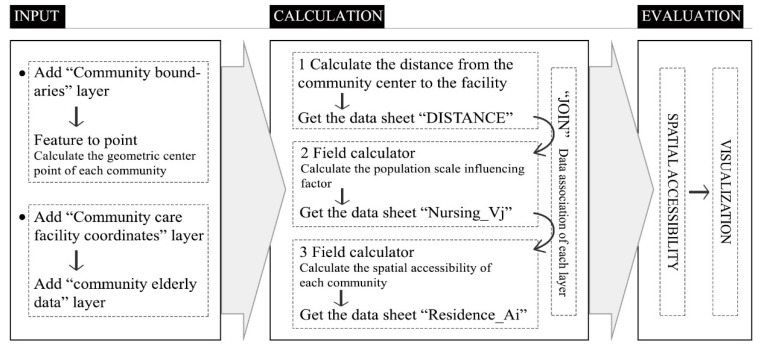
Implementation of the potential model in GIS.

**Figure 5 ijerph-17-05086-f005:**
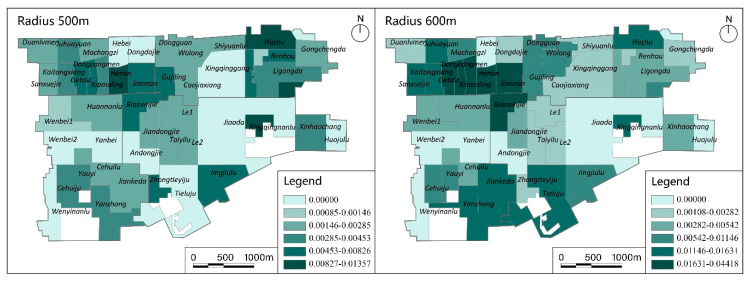
Calculation results of the 2SFCA model with search radii of 500 m (**left**) and 600 m (**right**).

**Figure 6 ijerph-17-05086-f006:**
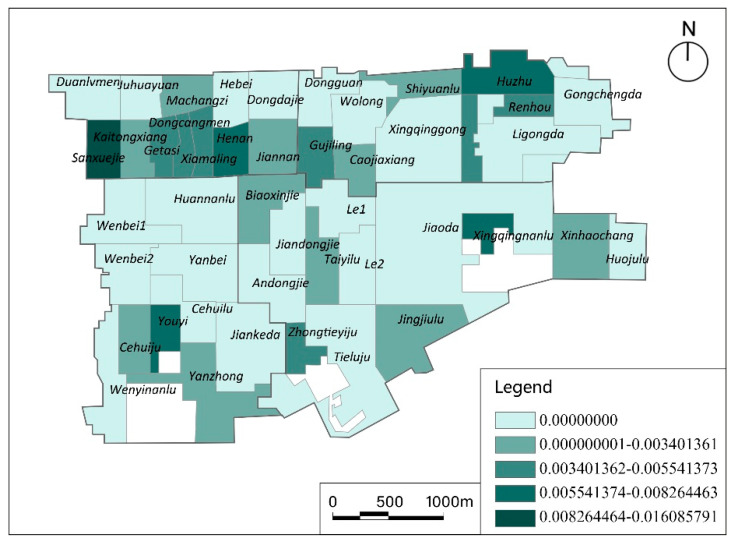
Calculation results of the potential model.

**Figure 7 ijerph-17-05086-f007:**
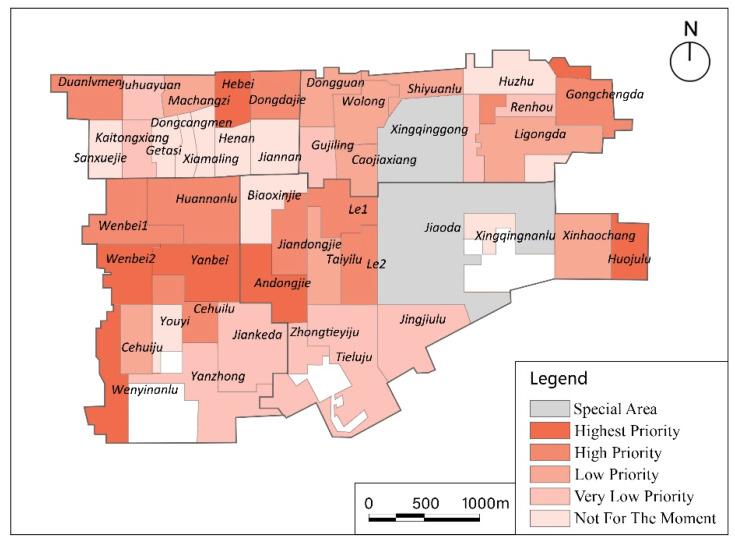
Suggestions of priority areas to increase the number of community care facilities or beds.

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
