# Peer review of "Two-Step Floating Catchment Area Model-Based Evaluation of Community Care Facilities’ Spatial Accessibility in Xi’an, China"

_ijerph, 2020, doi:10.3390/ijerph17145086_

Round 1
Reviewer 1 Report
1. According to current researches, the author selected 500 and 600m as the radius of community care facilities space accessibility, Is the distance reasonable for the study area? What are the areas of change in the two figures of Fig.5? Has the authors found a better value as radius through comparison? 2. What is the meaning of numerical expression in Fig. 5 and 6? Is there a scope for model values? 3. If the radius is very important to the author's research, it is not appropriate to use the community as the research scale. The author should take the raster scale as the research area, and the raster precision should be high. 4. The conclusion is not convincing. For example, what is the accuracy of the author's conclusion, how to verify it, and what is the sensitivity? How to deal with the boundary problems in author's research? 5. Other problems: 6. The map in Figure 1 is not standardized, and China's administrative region lacks important components. 7. in line 250, the value 55% should be 45%Author Response
Please see the attachment.

Reviewer 2 Report
This submitted paper has a clear focus on assessing the distribution of Xi'an community care facilities' spatial accessibility as well further the understanding of spatial allocation of urban community care facilities resources. Its broader scientific topic is "Spatial Accessibility" and the specific application is: "Community care facilities". It falls in the scope of "International Journal of Environmental Research and Public Health".
The paper has nice introduction with distinct sub-sections leading to a very gentle introduction to the topic and the problem. The authors show a very good understanding of the topic. Although, it has some potential for improvement:
- The general description of the problem and the description of its importance for the science and the society could be further improved by adding a bit of text regarding importance of this topic.
- A bit more text regarding the originality of this work and why it contains new results that significantly advance the research field.
- Could the results be more satisfactory if you have changed something in the methodology? Please add one/two sentences regarding soundness of the results.
- Are the results sensitive to this specific study area? I believe that the results are not sensitive. But, adding one-two sentences, about sensitivity would further improve the quality of the results.
- The "Conclusions" section is OK. Nevertheless, it could be further improved by describing the importance of this work as well as highlight clearly of potential further development of this methodology.
Overall this work is very good and it will be a nice addition to the next journal issue.
Reviewer 3 Report
The paper ‘Two-Step Floating Catchment Area Model-Based Evaluation of Community Care Facilities’ deals with an interesting topic related to elderly urban population and community care facilities. Two approaches (two-step floating catchment area method and potential model) are used by Authors to carry out scientific evaluation of the spatial accessibility of community care facilities in Beilin district of Xi’an. The aims of the paper are to explore best quantitative research methods for assessing the distribution of Xi’an community care facilities’ spatial accessibility, provide ideas for similar studies in the future, and further understanding of spatial allocation of urban community care facilities resources.
I found the paper well written and clearly developed. Maps and figures are very clear and useful. References are rich. I believe that the paper is near to publication after some minor revisions. Here follows some suggestions that could help Authors to improve the quality the paper:
- Aging is essentially a demographic process grounded in demographic transition theory (Thompson 1929; Chesnais 1992). I suggest to Author to recall it in the introduction section.
- It is not clear to me the sources of data on population and their time reference
- The 2SFCA method is demonstrated to have some fundamental shortcomings as proved for example in McGrail and Humphreys (2009). I suggest the Author to cite this contribution and to better underlined limits and shortcomings of 2SFCA method and of the potential model as well.
- I suggest to improve the policy implications of the results achieved. What about the future? Maybe some information about demographic projections of aging process could be useful to better understand future scenario and policy needs.
Round 2
Reviewer 1 Report
line 121-122, line138, line 355 should add related literatures to support your viewpoints.
line 226 should delet the blank.
line 257 should point out which Figures you used.
line 362 should be 'shortcomings'.
There are still some language problems in the manuscript, I suggest the authors read and revise the manuscript carefully.
Author Response
We would like to appreciate all the comments from the reviewer, for the careful and constructive reviews. We have revised our manuscript based on the comments, which are detailed as follows.
line 121-122, line139-140, line 358, the related literatures were added.
line 227, the blank was deleted.
line 259, the Figure we used is figure 5 and it was added.
line 366, we are very sorry for our incorrect writing and it was corrected as 'shortcomings'.
Our manuscript is revised for English language proofreading in the MDPI system.
Thank you so much for these good comments.